# Bioabsorption of Subcutaneous Nanofibrous Scaffolds Influences the Engraftment and Function of Neonatal Porcine Islets

**DOI:** 10.3390/polym14061120

**Published:** 2022-03-11

**Authors:** Purushothaman Kuppan, Sandra Kelly, Karen Seeberger, Chelsea Castro, Mandy Rosko, Andrew R. Pepper, Gregory S. Korbutt

**Affiliations:** 1Alberta Diabetes Institute, University of Alberta, Edmonton, AB T6G 2E1, Canada; kuppan@ualberta.ca (P.K.); sankelly@ualberta.ca (S.K.); ks1@ualberta.ca (K.S.); ccastro@ualberta.ca (C.C.); rosko@ualberta.ca (M.R.); 2Department of Surgery, University of Alberta, Edmonton, AB T6G 2E1, Canada

**Keywords:** nanofibrous scaffolds, subcutaneous space, neonatal porcine islets, type-I-diabetes, transplantation

## Abstract

The subcutaneous space is currently being pursued as an alternative transplant site for ß-cell replacement therapies due to its retrievability, minimally invasive procedure and potential for graft imaging. However, implantation of ß-cells into an unmodified subcutaneous niche fails to reverse diabetes due to a lack of adequate blood supply. Herein, poly (ε-caprolactone) (PCL) and poly (lactic-co-glycolic acid) (PLGA) polymers were used to make scaffolds and were functionalized with peptides (RGD (Arginine-glycine-aspartate), VEGF (Vascular endothelial growth factor), laminin) or gelatin to augment engraftment. PCL, PCL + RGD + VEGF (PCL + R + V), PCL + RGD + Laminin (PCL + R + L), PLGA and PLGA + Gelatin (PLGA + G) scaffolds were implanted into the subcutaneous space of immunodeficient Rag mice. After four weeks, neonatal porcine islets (NPIs) were transplanted within the lumen of the scaffolds or under the kidney capsule (KC). Graft function was evaluated by blood glucose, serum porcine insulin, glucose tolerance tests, graft cellular insulin content and histologically. PLGA and PLGA + G scaffold recipients achieved significantly superior euglycemia rates (86% and 100%, respectively) compared to PCL scaffold recipients (0% euglycemic) (* *p* < 0.05, ** *p* < 0.01, respectively). PLGA scaffolds exhibited superior glucose tolerance (* *p* < 0.05) and serum porcine insulin secretion (* *p* < 0.05) compared to PCL scaffolds. Functionalized PLGA + G scaffold recipients exhibited higher total cellular insulin contents compared to PLGA-only recipients (* *p* < 0.05). This study demonstrates that the bioabsorption of PLGA-based fibrous scaffolds is a key factor that facilitates the function of NPIs transplanted subcutaneously in diabetic mice.

## 1. Introduction

Islet transplantation re-establishes glucose homeostasis, eliminates hypoglycemic unawareness, improves glycosylated hemoglobin (HbA1c) levels and stabilizes secondary complications associated with type 1 diabetes (T1D) [1,2,3,4,5]. The limited supply of cadaveric human donors islets [6] and heterogenicity in human islet quality are major obstacles restricting islet transplantation to become the standard of care for all patients living with T1D. There is strong rationale to pursue the use of porcine donors for clinical islet xenotransplantation, including (a) the reproducibility and quality of preparing porcine islets by eluding co-morbidities, brain death insults and ischemic damage associated with human islet procurement [7,8,9]; (b) the ubiquitous availability of porcine islets; thereby increasing access and reducing transplant wait times [8]; (c) allowance of genetic modification and cloning of pigs to reduce immunogenicity [9,10,11]; and (d) porcine islets are a potential therapy for highly allosensitized patients [12]. In addition, our group previously reported a simple, reproducible, scalable and economical method to isolate large numbers of neonatal porcine islets (NPIs) for clinical use [8]. We have previously demonstrated that these NPIs, isolated from 1- to 3-day-old pancreata, can reverse hyperglycemia in mice [13], allogeneic pigs [14] and non-human primates [15,16].

Apart from the potential benefits, there are possible inherent risks and concerns associated with clinical NPI transplantation. These hurdles include ethical, moral and social concerns, hyperacute immunological responses and risk of zoonotic disease transmission [17]. Perception of NPI transplantation has historically been plagued by the possible transmission of porcine endogenous retro viruses (PERV). However, a recent clinical transplantation report of encapsulated NPIs stated that there was no evidence of PERV infection in all patients up to 600 days of post-transplant follow-up [18,19]. Seven-year follow-up data of thirty-eight patients with T1D transplanted with encapsulated porcine islets indicated no transmission of zoonotic diseases [20]. Most recently, the first successful attempt of clinical-grade porcine kidneys from genetically modified pigs was transplanted in a human decedent model that showed no observations of hyperacute rejection or transmission of porcine retroviruses [21]. Furthermore, surgeons from the University of Maryland School of Medicine successfully transplanted a genetically modified pig’s heart into the first human patient [22]. These recent acute studies demonstrate that genetic modifications of pigs may reduce some inherent risks, in the short-term, while demonstrating compatibility and functionality of xenografts. While exceedingly promising, long-term monitoring is required to effectively gauge the safety and efficacy of genetically modified pigs. The widespread use of porcine islets could be possible if a safe and recoverable ectopic transplant site for genetically modified pig islets is identified. This innovation could certainly advance the field of NPI clinical transplantation as an efficacious alternative treatment to human islet transplantation.

Transplantation of islets into the portal vein routinely establishes transient insulin independence; however, it has been associated with life-threating intraperitoneal bleeding, portal vein thrombosis and hepatic steatosis [23,24]. In addition to evoking a deleterious acute inflammatory response in which nearly 70% of the islet mass is destroyed, the liver may also contribute to gradual islet graft attrition [23]. Identification of an alternative and safer site for islet transplantation is therefore desirable [25]. The subcutaneous space is an attractive surrogate intrahepatic engraftment of stem cell-derived and xenogeneic islet transplantation due to its retrievability, accommodation of large transplant volumes, potential for monitoring cellular transplant function and localized drug delivery [26,27,28,29,30,31]. However, transplantation of islets into the native subcutaneous environment fails to reverse diabetes due to innate hypovascularity [32].

To create a retrievable subcutaneous site that supports the long-term function and survival of ß-cell grafts we utilized porous scaffolds fabricated from either poly(ε-caprolactone) (PCL) or poly(lactic-co-glycolic acid) (PLGA). PCL and PLGA are United States Food and Drug Administration (FDA)-approved biodegradable and biocompatible polymers that exhibit attractive and tunable physical properties [33]. Therefore, in this study we used PCL and PLGA as a backbone to construct fibrous scaffolds using electrospinning technology. Further, PCL fibrous scaffolds were functionalized by covalently binding a combination of the cell adhesion peptide (RGDC; Arginine-glycine-aspartate-cysteine), vascular endothelial growth factor (VEGF) peptide mimics, QK peptide, KLTWQELYQLKYKG) and extracellular matrix protein mimics (laminin peptide mimics, IKLLI). In addition, PLGA scaffolds were functionalized by being physically blended with gelatin, as gelatin will allow PLGA to be rapidly absorbed by host tissues [34] and the RGD motif in the gelatin will enhance islet cell adhesion [35]. In this study, we compared the efficacy of PCL and PLGA-based fibrous scaffolds with NPI, a clinically relevant extrahepatic subcutaneous site, in a mouse xenograft model.

## 2. Materials and Methods

### 2.1. PCL and PLGA Fibrous Scaffold Fabrication

Poly (ε-caprolactone) (PCL: Mw 80,000, Sigma–Aldrich, Oakville, ON, Canada) and poly(lactic-co-glycolic acid) (PLGA: Mw 38,000–54,000; 50:50 ratio, Sigma–Aldrich, Oakville, ON, Canada) fibrous scaffolds were prepared through electrospinning (Dual pump nanofiber electrospinning unit, Model # HO-NFES-043C, Holmarc, Kerala, India). The PCL scaffolds were functionalized with arginine–glycine–aspartate–cysteine (RGDC, Rasathantra Nanotech Innovations Inc, Edmonton, Canada) + vascular endothelial growth factor peptide mimics (VEGF) (PCL + R + V) and RGDC + Laminin peptide mimics (PCL + R + L) using our previously published protocol [36]. Both VEGF and laminin peptide mimics were synthesized using a microwave-assisted peptide synthesizer [36]. In brief, PCL electrospun fibrous scaffolds were produced at 10 kV with a flow rate of 0.05 mL/minute and were surface functionalized with RGDC through the bi-functional cross-linker sulfosuccinimidyl 4-(N-maleimidomethyl) cyclohexane-1-carboxylate covalently. Subsequently, RGDC-functionalized PCL scaffolds were further cross-linked with VEGF or laminin using ultraviolet light to obtain PCL + R + V and PCL + R + L scaffolds [36]. To prepare PLGA scaffolds, a PLGA polymer solution (18%, *w*/*v*) prepared in hexafluoroisopropanol (HFIP, Sigma–Aldrich, Oakville, ON, Canada) was dispensed using a syringe pump with a flow rate of 0.3 mL/hour, and a nanofiber jet stream was produced at 12 kV and collected in the rotating mandrel (150 mm diameter, 500 rpm). A thin fibrous PLGA sheet was removed from the mandrel and preserved in a vacuum desiccator prior to conducting the subsequent transplant studies. Similarly, PLGA + Gelatin (PLGA + G) thin fibrous sheets were produced by blending PLGA (18% *w*/*v*) and gelatin (10% *w*/*v*) solutions in a 5:5 ratio, with 12 kV applied potential and a 0.5 mL/hour flow rate. Nanofibrous scaffolds were characterized for fiber diameter, pore size and thickness. The fiber diameter and pore size of the scaffolds were measured by analyzing scanning electron micrographs, using ImageJ software (downloaded from the National Institutes of Health website (http://rsb.info.nih.gov/ij), accessed on 2 March 2022). Scaffold thickness was measured using a Vernier caliper (Starrett^®^ 799, model # 799A-8/200, Athol, MA, USA).

### 2.2. NPI Isolation, Transplantation and Metabolic Follow-Up

#### 2.2.1. NPI Isolation

All animal studies were conducted in accordance with the guidelines of the institutional ethical committee of University of Alberta and the Canadian Council of Animal Care. Porcine pancreases were surgically obtained from 1- to 3-day-old male and female Duroc neonatal piglets (University of Alberta Swine Research Centre, Edmonton, AB, Canada). Neonatal porcine islets (NPIs) were isolated and cultured for seven days as described previously [8,37]. A total of eight independent NPI isolations were used for the transplant studies.

#### 2.2.2. Scaffold Implantation, Transplantation and Metabolic Follow-Up

We first assessed the bio-absorption and biocompatibility of the scaffolds alone (no NPIs). To assist in the surgical subcutaneous implantation of the scaffolds and to create a lumen for subsequent NPI infusion, each scaffold was rolled around a nylon catheter (Adroit guiding catheter, JL3, 6F, 1.8 cm in length, Cordis, FL, USA) and implanted into the abdominal subcutaneous space of naïve male immune-compromised B6.129S7-Rag1^tm1Mom^/J (Rag) mice (*n* = 2 for each scaffold) (Jackson Laboratories) as well as naïve male immune-competent mice C57BL/6 (*n* = 2 for each scaffold) (Jackson Laboratories). After four weeks in situ, grafts were excised and histologically assessed.

For the NPI transplant studies, male and female immunodeficient B6.129S7-Rag1^tm1Mom^/J mice were used as recipients. For preparation of the subcutaneous site [38,39], PCL (*n* = 5), PCL + R + V (*n* = 3), PCL + R + L (*n* = 3), PLGA (*n* = 7) and PLGA + G (*n* = 7) catheter-rolled scaffolds were implanted subcutaneously in non-diabetic Rag mice. After four weeks, these mice were rendered diabetic by a single intraperitoneal (i.p) injection of 180 mg/kg streptozotocin freshly dissolved in acetate buffer (Sigma–Aldrich, Oakville, ON, Canada) seven days before transplantation. Controls included diabetic Rag mice implanted under the kidney capsule (KC) with NPI. Blood samples for glucose measurements were obtained from the tail vein (OneTouch UltraMini glucose meter). Mice with blood glucose greater than 18 mmol/L for two consecutive days were considered diabetic. A full mass of 3000 NPIs were transplanted under the KC or within the subcutaneous space.

All transplanted mice were monitored for weekly non-fasting blood glucose levels. When non-fasting blood glucose levels were ≤11.1 mmol/L, mice were considered normalized. Serum samples were collected for the measurement of graft-specific circulating porcine insulin at 10 and 20 weeks post-transplant. Basal (0 min, fasting) and stimulated (post i.p administration of 3 g/kg of 50% dextrose; 60 min) porcine insulin levels were measured in the recipient’s serum using ALPCO ultra-sensitive ELISA (ALPCO, Salem, NH, USA) according to the manufacturer’s protocol. An intra-peritoneal glucose tolerance test (IPGTT) was performed on transplanted mice at 20 weeks post-transplant. Mice were fasted overnight followed by i.p. administration with dextrose (3 g/kg), and blood glucose was measured at 0, 15, 30, 60, 90 and 120 min. Graft-dependent euglycemia was confirmed by survival nephrectomy of the graft-bearing kidney or subcutaneous graft excision at 27 weeks post-transplant. These animals were monitored for 48 h to confirm the recurrence of hyperglycemia and were subsequently euthanized. Recovered grafts underwent morphological analysis or were assessed for the cellular insulin content.

### 2.3. Characterization of Grafts

Scaffold-only grafts (no NPI) recovered at four weeks were fixed in 10% *w*/*v* zinc-buffered formalin (Thermo Fisher, Waltham, MA, USA), embedded in paraffin blocks and subsequently stained using hematoxylin and eosin (H&E) [37,39]. Similarly, NPI grafts were processed and embedded in paraffin blocks for histological assessment. The presence of insulin positive cells in the grafts was assessed using the avidin–biotin complex (ABC) method with peroxidase and diaminobenzidine as the chromogen in accordance with a previously published method [13]. The graft total cellular insulin content was extracted in accordance with previously published methods [8,38] and was analysed using a MesoScale Mouse/Rat Insulin Kit (Meso Scale Diagnostics, Rockville, MD, USA) according to the manufacturer’s protocol.

### 2.4. Statistical Analysis

Results are represented as the mean ± standard error mean (SEM) of (*n*) independent experiments. Kaplan–Meier function survival curves were compared using the log-rank (Mantel–Cox) statistical analysis for the percent euglycemic rate. Statistical significance of differences between and within the groups was calculated with one-way ANOVA and Student’s unpaired *t*-test, respectively. A *p*-value of <0.05 was required to consider the results significant.

## 3. Results

### 3.1. Scaffold Bioabsorption and Biocompatibility

Following electro-spinning, scanning electron micrographs confirmed a continuous fibrous architecture of the PCL (Figure 1A), PLGA (Figure 1B) and PLGA + G (Figure 1C) scaffolds. The average fiber diameter of the PCL, PLGA and PLGA + G scaffolds was 1000 ± 60 nm, 420 ± 20 nm and 200 ± 2 0 nm, respectively. The thickness of the PCL, PLGA and PLGA + G sheets was 142 ± 11 μm, 51 ± 3 μm and 20 ± 2.0 μm, respectively. The average pore size of the PCL, PLGA and PLGA + G nanofibrous scaffolds was 21 ± 4 μm^2^, 38 ± 5 μm^2^ and 40 ± 0.6 μm^2^, respectively. To simplify subcutaneous implantation and allow creation of a luminal space for subsequent islet transplantation, these scaffolds were rolled around a nylon catheter to provide mechanical support (Figure 1). The scaffolds were implanted subcutaneously in naive Rag and C57BL/6 mice and were explanted at four weeks post-implant and stained for H&E (Figure 2A,B). Following macroscopical and histological assessment, the PCL scaffolds were still observed indicating poor bioabsorption into the host tissue (Figure 2A1–A3,B1–B3). In contrast, the macroscopical presence of the PLGA scaffolds was not observed, and when examined histologically, the PLGA scaffolds were shown to be highly cellularized and absorbed within host tissues (Figure 2A4,A5,B4,B5). Furthermore, neovascularization was observed in and around the PLGA and PLGA + G scaffolds (Figure 2). In C57BL/6 mice, we did not observe adverse foreign body reactions, foreign body giant cells or mononuclear cell infiltration in or around either the PCL- or PLGA-based scaffolds (Figure 2B1–B5).

### 3.2. Metabolic Follow-Up of Transplant Recipients

Metabolic follow-up of non-fasting glycemia was measured on the transplanted mice weekly, and an intraperitoneal glucose tolerance test was administered once recipients reached and maintained euglycemia. Weekly non-fasting blood glucose levels of the KC, PLGA and PLGA + G groups were comparable (*p* > 0.05; Figure 3A). Eighty-six percent (6/7) of KC recipients, 86% (6/7) of the PLGA group and 100% (7/7) of the PLGA + G recipients achieved euglycemia (Figure 3B). In contrast, none of the animals attained normoglycemia in the PCL, PCL + R + V and PCL + R + L groups (Figure 3A). In addition, 5/5 of the mice in the PCL-only group were electively euthanized at two weeks post-transplant due to poor health conditions. At the endpoint of the study, 27-weeks post-transplant, the kidney-bearing NPI grafts and subcutaneous PLGA and PLGA + G grafts were removed. All animals returned to a diabetic state within 48 h (>18 mmol/L), confirming that the transplanted grafts were responsible for achieving euglycemia (Figure 3A). The euglycemic rate of the KC, PLGA and PLGA + G groups was statistically significant when compared with the PCL, PCL + R + V and PCL + R + L groups (* *p* < 0.05, ** *p* < 0.01). The median time to normoglycemia for the KC, PLGA and PLGA + Gelatin groups was comparable (17.3 ± 3.2, 21.4 ± 2.8 and 17.4 ± 3.3 weeks, respectively) (*p* > 0.05).

Glycemic responses to a metabolic challenge were comparable between the KC, PLGA and PLGA + G recipients (Figure 4A,B). In contrast, recipients of PCL + R + V and PCL + R + L were glucose intolerant compared the recipients of KC, PLGA and PLGA + G (* *p* < 0.05) (Figure 4A,B). Serum samples were analyzed for porcine insulin following the administration of an intraperitoneal bolus of glucose (3 g/kg) at 0 min (basal) and 60 min post-glucose administration (stimulated). At 10 weeks post-transplant, stimulated porcine insulin levels vs. basal levels were significantly higher in recipients transplanted with NPI in the subcutaneous space of PLGA (** *p* < 0.01) and PLGA + G (** *p* < 0.01) (Figure 4C). In contrast, KC, PCL + R + V and PCL + R + L recipients were not significantly different between the basal and stimulated porcine insulin levels (Figure 4C; *p* > 0.05) at 10 weeks. At 20 weeks post-transplant, the KC (*** *p* < 0.001), PLGA (* *p* < 0.05) and PLGA + Gelatin (** *p* < 0.01) groups exhibited significantly higher stimulated porcine insulin levels vs. basal levels, whereas the PCL + R + V and PCL + R + L recipients did not (*p* > 0.05; Figure 4E). The calculated stimulated porcine insulin secretion index of KC recipients was statistically significant between the groups except the PLGA + G groups at 20 weeks (Figure 4F) (* *p* < 0.05, ** *p* < 0.01).

### 3.3. Graft Morphological Characterization and Cellular Insulin Content of Grafts

Gross morphological examination of the grafts at 27 weeks post-transplantation revealed new blood vessel formation in and around the grafts (Figure 5A). The PCL, PCL + R + V and PCL + R + L scaffolds (Figure 5A2–A4) were shown to be present at the implantation site, indicating that these PCL scaffolds were not bioabsorbed into the host tissue. In contrast, the PLGA and PLGA + G scaffolds were not observed and were largely absorbed by the host tissue (Figure 5A5,A6). Immuno-histochemical characterization of NPI grafts revealed the presence of numerous intact islets with robust insulin positive cells under the KC (Figure 5B1) or within the subcutaneous PLGA (Figure 5B5) and PLGA + G (Figure 5B6). In contrast, few intact insulin positive islets were detected in the PCL (Figure 5B2), PCL + R + V (Figure 5B3) and PCL + R + L grafts (Figure 5B4). The total cellular insulin content of the NPI grafts in the KC (44.1 ± 6.5 μg) and PLGA + G (48.6 ± 4.7 μg) scaffold was comparable (*p* > 0.05) (Figure 6). In contrast, NPI grafts in the PLGA + G scaffold had a higher total cellular insulin content than the PLGA scaffold (28.1 ± 3.9 μg) (1.7 fold, * *p* < 0.05).

## 4. Discussion

The subcutaneous space has been shown to be an alternative transplant site for islet transplantation; however, this transplant site requires suitable modifications to facilitate islet engraftment. Islets transplanted into the unmodified subcutaneous space resulted in primary non-function and fail to correct the diabetes [40]. This was mainly due to the islets in an unmodified space experiencing hypoxic conditions, resulting in apoptosis [41]. Our group recently showed that NPI transplanted without a fibrin scaffold in subcutaneous space did not achieve optimal graft function [37]. Therefore, preconditioning of the subcutaneous space has been pursued.

Biomaterial scaffolds are an appealing and encouraging approach to modify the subcutaneous space into a suitable microenvironment to facilitate long-term islet engraftment and functions. Biomaterial scaffolds combined with proangiogenic, anti-inflammatory or immunomodulatory factors have been demonstrated to expedite and enhance neovascularization and rapid tissue ingrowth and to improve the exchange of oxygen, nutrients, metabolic products and waste removal [42,43,44]. However, bioabsorption of the materials needs to be tailored to obtain optimal host integration and islet engraftment. Herein, we used biodegradable and biocompatible materials to make fibrous PCL and PLGA scaffolds to create a subcutaneous transplant site for beta cell replacement therapies. These two materials are FDA approved and are widely used polymer systems in biomedical devices such as tissue-engineered grafts, drug eluting particles, sutures, implants and prosthetic devices [45,46].

Since, native PCL and PLGA polymers do not contain natural cell recognition motifs such as RGD [34,47], we functionalized these polymers with peptide mimics such as cell-adhesion peptide RGD, VEGF-mimicking QK peptide and laminin-mimicking IKLLI peptide. For the PCL fibrous scaffold, functionalized was achieved via ultraviolet cross-linking; whereas for PLGA, a physical method blended with gelatin (partially hydrolysed collagen that preserves the cell adhesion of the RGD motif) was performed, which enables better bioabsorption, superior cell adhesion and subsequent vascularization compared to PLGA and PCL [34,48,49].

The importance of extracellular matrix (ECM) components such as collagen, laminin and fibronectin for islet function has been demonstrated, signifying their critical role in organizing networks between endocrine cells, vascular endothelial cells, neural cells and immune cells [50,51,52,53]. These interactions facilitate the rapid signalling between the islet graft and host tissues, which enables graft survival, viability, restored insulin secretion and tissue remodelling [50,54]. During the enzymatic and mechanical digestion process inherent in islet isolation, the native ECM and the vasculature are disrupted [55], thus leading to a decrease in cell viability and suboptimal transplant outcomes [56]. Hence, impregnating scaffolds with ECM components may be essential to recapitulate the endogenous islet microenvironment heterotopically.

Preconditioning techniques to modify the subcutaneous site have included basic fibroblast growth factor-impregnated gelatin microspheres, biocompatible devices consisting of a cylindrical stainless steel mesh or methacrylic acid-coated silicone tube or temporary implanted catheters [39,57,58,59]. All of these studies have shown variable functional outcomes. Hence, in this study, we explored the role of functionalized PCL and PLGA fibrous scaffolds in islet xenograft function at a clinically relevant subcutaneous transplant site. It is well known that a material’s biodegradation and bioabsorption properties are crucial for tissue regeneration and remodelling [60]. To acquiesce tissue ingrowth, the scaffold materials have to biodegrade. In this study, we observed that PCL-based fibrous scaffolds were not absorbed by the host milieu whereas the PLGA-based fibrous scaffolds were completely absorbed by the host tissues at the same time post-implant. In addition, we found the presence of intact PCL-based scaffolds at the transplant site when they were excised for histological analysis at the study endpoint (~31 weeks post-implantation). The non-absorption of PCL material appears to evoke a physical barrier between the islets and host tissues, drastically affecting their engraftment, viability and function. Hence, none of the recipients in the PCL-based groups achieved euglycemia. In contrast, animals from the PLGA + G and PLGA groups achieved 100% and 86% euglycemia, respectively. We observed rapid graft function in PLGA + G recipients, 11 weeks post-transplant, whereas to achieve a similar percent of euglycemia, PLGA recipients took approximately 22 weeks post-transplant. Furthermore, we observed a significant amount of total cellular insulin content in the PLGA + G grafts compared to the PLGA grafts. These observations indicate the significance of PLGA functionalization with gelatin and suggest that the degree of bioabsorption of fibrous scaffolds (PLGA vs. PCL scaffolds) is essential to achieve islet graft function within the subcutaneous space.

We speculate that the inability of our PCL-based scaffolds to bioabsorb could be due to the physical properties of the PCL materials and scaffold processing conditions, including (i) the higher molecular weight of the PCL (Mw: 80,000 g/mol) employed for the scaffold fabrication, (ii) the inherent hydrophobic nature of the PCL material, (iii) the scaffold-functionalization processing conditions (UV cross-linking employed for peptide conjugation to the PCL scaffolds) and (iv) the thickness of the PCL fibrous sheet. Our observations align with previous studies, which have shown that electrospun fibrous PCL had absorbed 27% of its molecular weight 90 days post subcutaneous implantation in rats [61]. Similarly, Sun et al., demonstrated that PCL capsules remained intact in the subcutaneous space up to two years post-implantation in rats [62]. Our previous studies demonstrate that PLGA drug-eluting micelles completely disappeared by 60 days post-implantation [63]. Similarly, in the present study, we observed rapid bioabsorption of fibrous PLGA and PLGA + G scaffolds.

We anticipated that impregnating the PCL scaffold with VEGF and laminin peptide mimics would enhance the islet graft function. However, to the contrary, we observed poor graft function with all PCL-based scaffolds compared to PLGA-based scaffolds. PLGA is less hydrophobic compared to the PCL and moreover has a lower molecular weight. Furthermore, gelatin is a hydrophilic material that enhances PLGA degradation, while the RGD motif in the gelatin aids in cell adhesion and recruitment. Factors such as the hydrophobicity, molecular weight and chemical structure of polymers play important roles in the biodegradation process [64]. In addition, hydrophilic surfaces are beneficial for the adhesion, proliferation and growth of cells [34]. Collectively, our observations of rapid bioabsorption and subsequent islet graft function in PLGA + G scaffold recipients compared to PLGA and PCL scaffold recipients further provide insight into the dynamic influence the physical and chemical properties of biomaterials have on host interactions and in our hands, cell therapy efficacy.

## 5. Conclusions

Taken together, electrospun PCL and PLGA fibrous scaffolds possess attractive material properties to support islet transplant function in the extrahepatic subcutaneous space. We demonstrated that subcutaneously implanted PLGA and PLGA + G scaffolds provided an appropriate microenvironment and prompt vascularization, which facilitated long-term islet graft function and diabetes correction in mice. Our study highlights the necessity of complete bioabsorption of nanofibrous, electrospun scaffolds to achieve NPI engraftment and graft function when transplanted into the subcutaneous space. Therefore, functionalized PLGA-based fibrous scaffolds warrant further investigation in preclinical large animal models for future clinical applications.

## Figures and Tables

**Figure 1 polymers-14-01120-f001:**
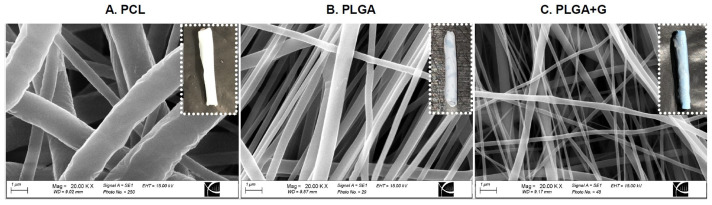
Scanning electron micrographs (SEM) demonstrate the fibrous architecture of electrospun PCL, PLGA and PLGA + G scaffolds. Fibers were continuous and randomly oriented. Macroscopic inserts illustrate the scaffolds rolled around the nylon catheter prior to subcutaneous implantation.

**Figure 2 polymers-14-01120-f002:**
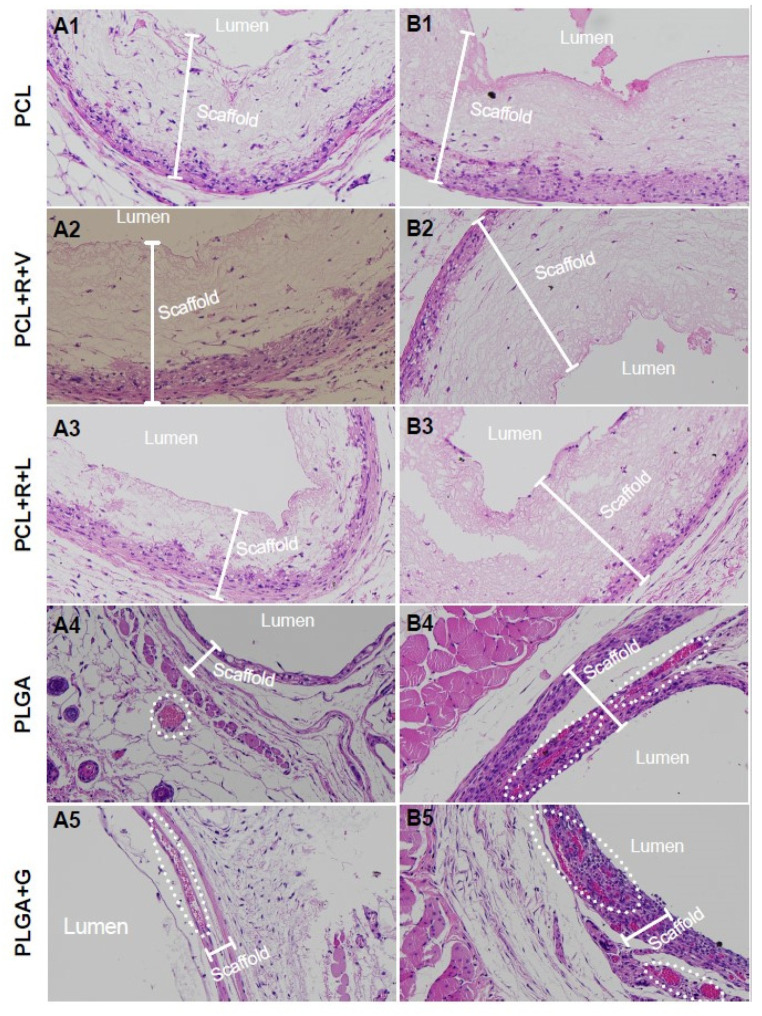
H&E staining of PCL (**A1**,**B1**), PCL + R + V (**A2**,**B2**), PCL + R + L (**A3**,**B3**), PLGA (**A4**,**B4**) and PLGA + G (**A5**,**B5**) scaffolds following four weeks post-implantation in the subcutaneous space of naïve B6.129S7-Rag1^tm1Mom/J^ (**A**) and C57BL/6 mice (**B**). PCL, PCL + R + V and PCL + R + L scaffolds were cellularized predominately at the outer surface of the scaffold, whereas the PLGA and PLGA + G scaffolds were more thoroughly cellularized and integrated with host tissues. Double-headed arrows indicate the scaffold, and dotted circles indicate neovascularization.

**Figure 3 polymers-14-01120-f003:**
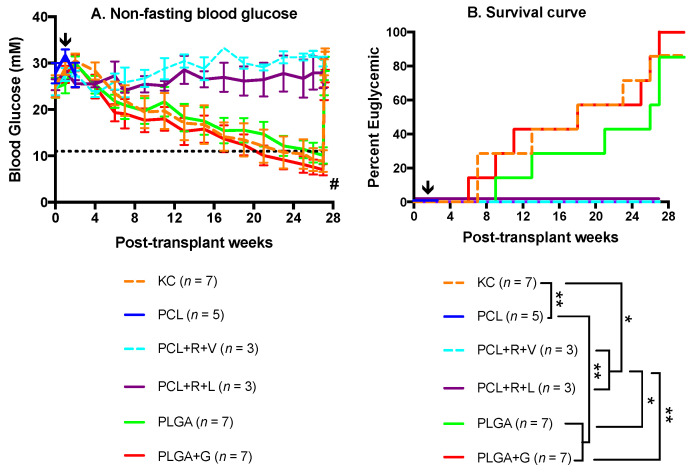
Metabolic follow-up of diabetic B6.129S7-Rag1^tm1Mom/J^–immunodeficient mice transplanted with neonatal porcine islets (3000 NPIs) under the KC or subcutaneously with PCL, PCL + R + V, PCL + R + L, PLGA or PLGA + G scaffolds. KC transplant recipients were included as a positive control. Mice were monitored for weekly non-fasting blood glucose (**A**) and were analyzed for their percent euglycemic (**B**). PLGA + G (7/7) recipients achieved euglycemia while PLGA (6/7) and KC (6/7) recipients achieved euglycemia (*p* > 0.05, log-rank). In contrast, none of the recipients normalized in PCL (0/5), PCL + R + V (0/3) or PCL + R + L (0/3) (* *p* < 0.05, ** *p* < 0.01, log-rank). # indicates graft retrieved, at which point mice reverted back to their pre-transplant hyperglycemic level in the non-fasting blood glucose graph, indicating survival nephrectomy or subcutaneous graft explantation. Arrows indicate that recipients of the PCL group (*n* = 5) were electively euthanized at two weeks post-transplant due to poor health conditions.

**Figure 4 polymers-14-01120-f004:**
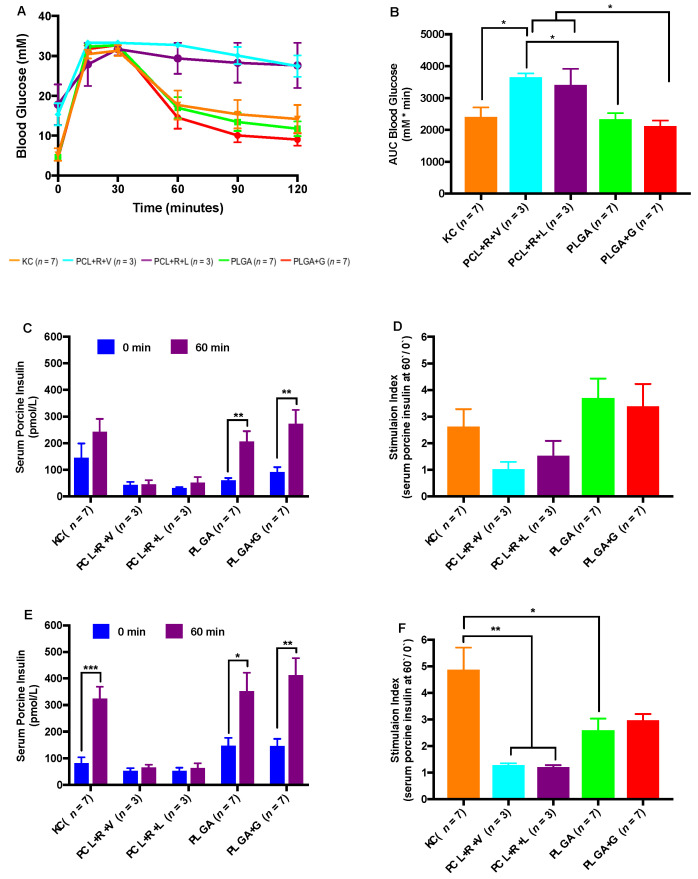
Intraperitoneal glucose challenge and stimulated porcine insulin secretion post-transplant. Transplanted recipients were subjected to an intraperitoneal glucose tolerance test (IPGTT) at 20 weeks post-transplant. Blood glucose during an IPGTT (**A**) and area under the curve for respective IPGTT ((**B**) * *p* < 0.05, one-way ANOVA). Porcine-stimulated serum insulin secretion prior to (0 min) and following 60 min after an i.p. injection of a glucose bolus was measured at 10 ((**C**) ** *p* < 0.01, two-tailed unpaired *t*-test) and 20 ((**E**) * *p* < 0.05, ** *p* < 0.01, *** *p* < 0.001, two-tailed unpaired *t*-test) weeks post-transplant. Calculated stimulation index for porcine-stimulated insulin secretion at 10 weeks (**D**) and 20 weeks post-transplant ((**F**) * *p* < 0.05, ** *p* < 0.01, one-way ANOVA).

**Figure 5 polymers-14-01120-f005:**
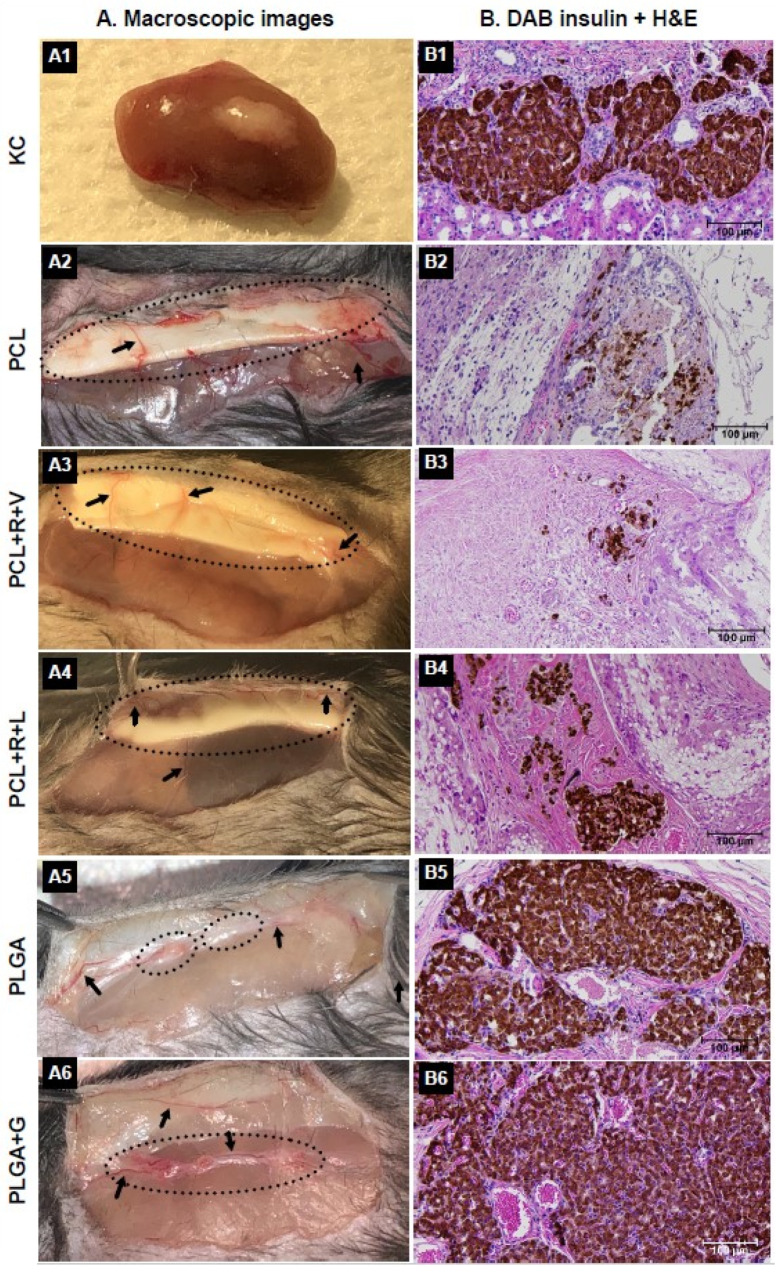
Macroscopic images of KC (**A1**), PCL (**A2**), PCL + R + V (**A3**), PCL + R + L (**A4**), PLGA (**A5**) and PLGA + G (**A6**) grafts at 27 weeks post-transplant. PCL + R + V and PCL + R + L scaffolds were intact and were not absorbed even at 31 weeks post-implantation whereas PLGA and PLGA + G scaffolds disappeared completely (indicated by circle). Interestingly, blood vessel innervations (indicated by arrows) were observed in the subcutaneous scaffolds. Of note, enormous blood vessels were observed in the PLGA and PLGA + G scaffolds. Immunostaining of grafts confirmed the presence of intact islet clusters with robust insulin positive cells (brown) in the KC (**B1**), or PLGA (**B5**) and PLGA + G (**B6**) scaffolds compared to few insulin positive islet clusters in the PCL (**B2**), PCL + R + V (**B3**) and PCL + R + L (**B4**) scaffolds. Of note, distinguishable blood vessels inside the grafts were seen in the PLGA and PLGA + G scaffolds.

**Figure 6 polymers-14-01120-f006:**
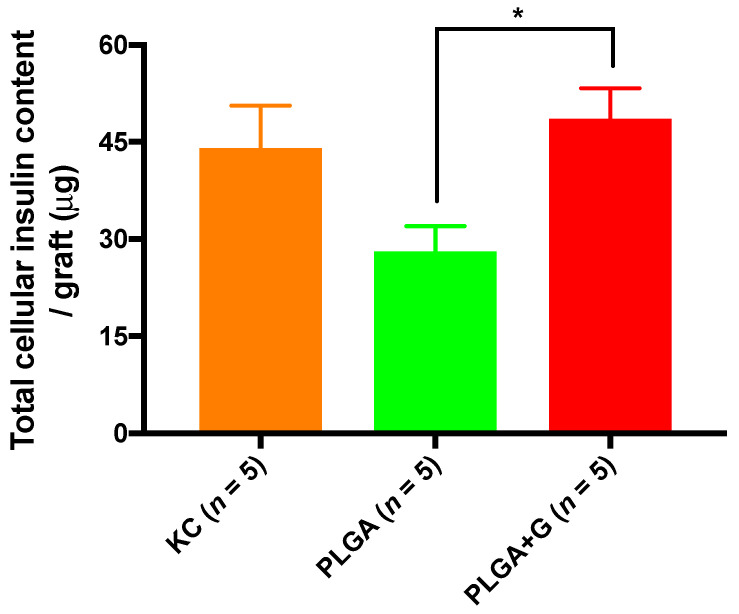
Total cellular insulin contents of the grafts at 27 weeks post-transplant. There were comparable insulin contents between the KC and PLGA + G scaffold (*p* > 0.05, one-way ANOVA). Of note, grafts from PLGA + G had higher total cellular insulin contents compared to the PLGA scaffold (* *p* < 0.05, one-way ANOVA).

## Data Availability

Not Applicable.

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
