# Peer review of "Bioabsorption of Subcutaneous Nanofibrous Scaffolds Influences the Engraftment and Function of Neonatal Porcine Islets"

_polymers, 2022, doi:10.3390/polym14061120_

Round 1

Reviewer 1 Report

The authors reported a pro-neovascularization bioabsorbable PLGA scaffold which facilitated long-term pig islet xenograft function and diabetes correction in mice. The data are quite promising. Some questions should be addressed before publication.

  1. Although this paper presents a strong argument for the importance of scaffold bioabsorption in long-term graft function and survival, the authors compare electrospun PCL and PLGA functionalized using different modifications. This introduces an inconsistency in the comparison between the PCL and PLGA scaffolds that is not explicitly addressed in the paper. While it is stated that PCL was functionalized with RDG, VEGF, and Laminin to mimic ECM, it is unclear why not gelatin was used to functionalize PCL, even though gelatin was used to functionalize PLGA through a seemingly simpler method.

  1. More characterizations are required for the electrospun fibrous scaffolds, such as pore size, fiber diameter, and scaffold thickness, which would have big effect on the mass transfer performance of the scaffold and related therapeutic function of the grafts.

  1. As descripted in the Methods section, each electrospun fibrous scaffold was rolled around a nylon catheter and then implanted together for the prevascularization at the subcutaneous site. However, the nylon catheter itself was also reported in a Device-Less method which can induce a robust neovascularization.

  1. The H&E staining Only is not enough to deliver the statement of the scaffold biocompatibility as below: “In the C57BL/6 mice we did not observe adverse foreign body reactions, foreign body giants cells or mononuclear cell infiltration in and around either the PCL or PLGA based scaffolds (Figure 2B1-B5), thereby indicating biocompatibility of the scaffolds.”(Page 5)

  1. Some experiment details were missing. Such as, how many pig islets (and purity) were transplanted into each mouse? And was this islet dose a full mass to correct diabetes?

  1. How come such a big delay (20 weeks after islet transplantation for over half of the recipients) was observed for the diabetes correction (i.e. BG < 11.1 mM)?

Author Response

Reviewer: 1

The authors reported a pro-neovascularization bioabsorbable PLGA scaffold which facilitated long-term pig islet xenograft function and diabetes correction in a mouse model. The data are quite promising. Some questions should be addressed before publication.

  1. Although this paper presents a strong argument for the importance of scaffold bioabsorption in long-term graft function and survival, the authors compare electrospun PCL and PLGA functionalized using different modifications. This introduces an inconsistency in the comparison between the PCL and PLGA scaffolds that is not explicitly addressed in the paper. While it is stated that PCL was functionalized with RDG, VEGF, and Laminin to mimic ECM, it is unclear why not gelatin was used to functionalize PCL, even though gelatin was used to functionalize PLGA through a seemingly simpler method.

Thank you for your valuable suggestions. By comparing the bioabsorption and functional efficacy of PCL and PLGA scaffolds alone (ie: no gelatin), we observed that PLGA scaffolds were completely bioabsorbed and 86% (6 out of 7) of the recipient mice became euglycemic post-transplant. In contrast, when PCL scaffolds alone (ie: no RGD, VEGF, or Laminin) were utilized all NPI recipients failed to exhibit a reversal of diabetes and had to be electively euthanized at 2 weeks post-transplant due to poor health conditions. In order to improve the metabolic outcome when using PCL scaffolds we sought to functionalized these polymers with RGD, VEGF, or Laminin. We however did not observe any improvement of NPI graft function in these now functionalized PCL recipients. This negative outcome could be due to the physical properties of PCL, such as (i) the higher molecular weight of the PCL (Mw: 80,000 g/mol) employed for the scaffold fabrication, (ii) inherent hydrophobic nature of the PCL material, and (iii) and the scaffold-functionalization processing conditions (UV cross-linking employed for peptide conjugation to the PCL scaffolds).

Therefore, we speculated that functionalization of PLGA with RGDC, VEGF and laminin via UV cross-linking would negatively impact the NPI graft function. Hence, we added gelatin to PLGA before electrospinning to produce a physically blended PLGA+G scaffold. Moreover, since gelatin will allow PLGA to be rapidly absorbed by host tissues and the RGD motif in the gelatin we further hypothesized that adding gelatin to PLGA will enhance NPI graft function. As we expected, we observed rapid correction of diabetes in PLGA+G recipients (11 weeks post-transplant), whereas PLGA alone recipients required approximately 22 weeks post-transplant to achieve 100% correction of diabetes. We agree that addition of gelatin into the PCL may improve bioabsorption, however the inherent hydrophobic nature and higher molecular weight of PCL would most likely negatively impact bioabsorption.          

  1. More characterizations are required for the electrospun fibrous scaffolds, such as pore size, fiber diameter, and scaffold thickness, which would have big effect on the mass transfer performance of the scaffold and related therapeutic function of the grafts.

Thank you for the suggestions regarding additional scaffold characterizations. The average fibers diameter of PCL, PLGA, and PLGA+G is 1000±60 nm, 420±20 nm, and 200±20 nm, respectively. The average thickness of the PCL, PLGA, and PLGA+G scaffolds is 142±11 μm, 51±3 μm, and 20±2.0 μm, respectively. The average pore size of PCL, PLGA and PLGA+G nanofibrous scaffolds is 21±4 μm2, 38±5 μm2 and 40±0.6 μm2, respectively. These data have now been included in the revised manuscript within the Results (Section 3.1).

  1. As descripted in the Methods section, each electrospun fibrous scaffold was rolled around a nylon catheter and then implanted together for the prevascularization at the subcutaneous site. However, the nylon catheter itself was also reported in a Device-Less method which can induce a robust neovascularization.

We agree that the nylon catheters alone can induce neovascularization. In the current study, the nylon catheters were completely covered by the scaffolds and therefore we expect minimal if any effect of the nylon catheters inducing neovascularization due to the physical barrier between the host tissue and the nylon biomaterial. In addition, since we utilized nylon catheters as temporary stents (to provide a lumen into which cells were infused upon catheter removal), in all scaffold groups (ie. PCL, PLGA and respective functionalized scaffolds), any influence the catheters may have on inducing neovascularization would be consistent between the scaffold groups.

  1. The H&E staining Only is not enough to deliver the statement of the scaffold biocompatibility as below: “In the C57BL/6 mice we did not observe adverse foreign body reactions, foreign body giants cells or mononuclear cell infiltration in and around either the PCL or PLGA based scaffolds (Figure 2B1-B5), thereby indicating biocompatibility of the scaffolds.”(Page 5)

Thank you for the comment. We agree and have rephrased this sentence in the revised manuscript: “In the C57BL/6 mice we did not observe foreign body giant cells or mononuclear cell infiltration in or around either the PCL or PLGA based scaffolds (Figure 2B1-B5)”.

  1. Some experiment details were missing. Such as, how many pig islets (and purity) were transplanted into each mouse? And was this islet dose a full mass to correct diabetes?

Thank you for your valuable suggestions. In this study, all grafts consisted of 3000 NPIs. We have included the following sentence in Section 2.2.2, “A full mass of 3000 NPI were transplanted under the KC or within the subcutaneous space”. As previously reported, NPIs are not fully differentiated like adult islets and they contain approximately 25% insulin-positive and 10% glucagon positive cells [JCI, 97(9), 1996, 2119-2129]. As reported in our previous studies this is the regular (full) dose we use to correct diabetes [Xenotransplantation, 28(3), 2021, e12669; Xenotransplantation, 28(6), 2021, e12706; Xenotransplantation, 27(4), 2020, e12581].    

  1. How come such a big delay (20 weeks after islet transplantation for over half of the recipients) was observed for the diabetes correction (i.e. BG < 11.1 mM)?

Since NPIs only contain approximately 25% insulin-positive ß-cells and have been shown to exhibit ß-cell proliferation post-transplant [JCI, 97(9), 1996, 2119-2129], we have routinely observed that it requires approximately 20 weeks post-transplant to achieve a sufficient ß-cells mass to correct diabetes in mice [Xenotransplantation, 28(3), 2021, e12669; Xenotransplantation, 28(6), 2021, e12706; Xenotransplantation, 27(4), 2020, e12581].

Reviewer 2 Report

Please correct syntax errors throughout the text, such as the following:

page 2-line 17: ‘a genetically modified pig’s into the’

page 2-line 23: ‘however, if a safe recoverable ectopic transplant site with genetically modified pigs is identified. It would certainly advance the field of NPI clinical transplantation as an efficacious alternative treatment for human islet transplantation.’

page 3-line 18: ‘prior to conduction the transplant the studies.’ please correct the syntax page 4: ‘Graft excised animals….followed by euthanized.’ ‘according to manufactures’ protocol.’ please correct

page 11: ‘into a more suitably microenvironment’

page 12: ‘which enabling the better bioabsorption’

page 13: ‘scaffold recipeints’

-briefly describe the electrospinning conditions for the PCL scaffolds

-provide SEM images of the functionalized PCL scaffolds, as well. What is the fibers’ diameter? Did you perform any additional characterizations on the fibers to confirm functionalization or proper blending in the case of PLGA-gelatin?

Author Response

Please correct syntax errors throughout the text, such as the following:

As suggested, we have carefully revised the manuscript to correct syntax errors thereby improving the manuscript’s readability. Specific comments have been addressed below.

Page 2-line 17: ‘a genetically modified pig’s into the’

Thank you for pointing out this error. It has been changed as “a genetically modified pig’s heart into the first human patient” in the revised manuscript.

Page 2-line 23: ‘however, if a safe recoverable ectopic transplant site with genetically modified pigs is identified. It would certainly advance the field of NPI clinical transplantation as an efficacious alternative treatment for human islet transplantation.

Thank you for the suggestion. It has been changed as “Widespread use of porcine islets could be possible if a safe and recoverable ectopic transplant site, for genetically modified pig islets, is identified. This innovation could certainly advance the field of NPI clinical transplantation as an efficacious alternative treatment to human islet transplantation.”

Page 3-line 18: ‘prior to conduction the transplant the studies.’ please correct the syntax page 4: ‘Graft excised animals….followed by euthanized.’ ‘according to manufactures’ protocol.’ please correct

Thank you for your valuable comments. We have revised the manuscript accordingly.

Page 11: ‘into a more suitably microenvironment’

Thanks for the suggestion. This sentence has been revised to “into a suitable microenvironment to facilitate long-term islet engraftment and functions” in the revised draft.

Page 12: ‘which enabling the better bioabsorption’

Thanks for the comment. This revised sentence now reads, “which enables better bioabsorption, superior cell adhesion and subsequent vascularization compared to PLGA and PCL” in the revised draft.

Page 13: ‘scaffold recipeints’

Thank you for your valuable suggestions. We have revised the manuscript accordingly.

-Briefly describe the electrospinning conditions for the PCL scaffolds

Now we have included the electrospinning process conditions for the PCL scaffolds in the revised manuscript (Methods Section 2.1).

-Provide SEM images of the functionalized PCL scaffolds, as well. What is the fibers’ diameter? Did you perform any additional characterizations on the fibers to confirm functionalization or proper blending in the case of PLGA-gelatin?

Thank you for the comments. The average fiber diameter of PCL, PLGA, and PLGA+G is 1000±60 nm, 420±20 nm, and 200±20 nm, respectively. Similarly, the thickness of the PCL, PLGA, and PLGA+G sheets is 142±11 μm, 51±3 μm, and 20±2.0 μm, respectively. We have included these characteristics in the revised manuscript (Section 3.1). We did not perform additional characterization on the fibers to confirm functionalization. However, we observed differences in graft function between the functionalized PCL and PLGA scaffolds suggesting that the importance of functionalization of scaffolds. We did not take SEM images from functionalized PCL scaffolds as we anticipated the functionalization of PCL scaffolds with RGD, VEGF and laminin would not change the fibrous architecture of the PCL scaffolds.